# The Effect of a New *N*-hetero Cycle Derivative on Behavior and Inflammation against the Background of Ischemic Stroke

**DOI:** 10.3390/molecules27175488

**Published:** 2022-08-26

**Authors:** Denis A. Borozdenko, Tatiana A. Shmigol, Aiarpi A. Ezdoglian, Darya I. Gonchar, Natalia. Y. Karpechenko, Dmitri N. Lyakhmun, Anastasia D. Shagina, Elvira A. Cherkashova, Daria D. Namestnikova, Ilya L. Gubskiy, Anastasia A. Chernysheva, Nina M. Kiseleva, Vadim V. Negrebetsky, Yuri I. Baukov

**Affiliations:** 1Department of Medicinal Chemistry and Toxicology, Pirogov National Research Medical University, Ministry of Health of Russia, 117997 Moscow, Russia; 2Department of Medical Nanobiotechnologies, Pirogov National Research Medical University, Ministry of Health of Russia, 117997 Moscow, Russia; 3Department of Neurology, Neurosurgery and Medical Genetics, Pirogov National Research Medical University, Ministry of Health of Russia, 117997 Moscow, Russia; 4Serbsky National Medical Research Center for Psychiatry and Narcology, 119034 Moscow, Russia

**Keywords:** neuroprotective activity, inflammation, transient MCAO, experimental ischemic stroke, behavioral tests

## Abstract

Ischemic stroke triggers a whole cascade of pathological changes in the brain, one of which is postischemic inflammation. Since in such cases thrombolytic therapy is often not possible, methods that modulate inflammation and affect microglia become particularly interesting. We synthesized 3-(2-oxo-4-phenylpyrrolidin-1-yl)propane-1-sulfonate calcium(II) (Compound **4**) and studied its anti-inflammatory activity in in vitro and in vivo models of inflammation and ischemia. Macrophage cell line RAW 264.7 was treated with lipopolysaccharides (LPS) and Compound **4** at various dosages to study the cytokine profile using real-time PCR and cytometric bead array (CBA). Stroke in rats was simulated by the middle cerebral artery occlusion method (MCAO). Several tests were performed to characterize the neurological deficit and locomotor activity of the rats, and afterwards, postmortem, the number of astrocytes was counted using immunohistochemistry. Compound **4** in in vitro tests dose-dependently reduced the expression of interleukin-1β (IL1β), and inducible nitric oxide synthase (iNOS) genes in cell culture and increased the concentration of cytokines: interleukin-2, 4, 6 (IL-2, IL-4, and IL-6). In vivo Compound **4** increased the orienting-exploratory behavior, and reduced neurological and motor deficit. The number of astrocytes that promote and support inflammation was lower in the group treated with Compound **4**. The stroke volume measured by magnetic resonance imaging (MRI) showed no difference. We have shown that Compound **4** demonstrates anti-inflammatory activity by increasing the synthesis of anti-inflammatory and reducing pro-inflammatory cytokines, and positively affects the neurological deficit in rats. Thus, Compound **4** has a high therapeutic potential in the management of patients after a stroke and requires further study of its neuroprotective properties.

## 1. Introduction

According to the World Health Organization (WHO), stroke is the second leading cause of death and the third leading cause of disability [1]. The number of stroke patients is steadily increasing as the life expectancy of the world’s population increases. The leading causes of disability in the postischemic period are neurological disorders, including dementia, locomotor insufficiency, and speech apparatus deficit. The mechanism of development of mnestic disorders is sophisticated: it consists of a violation of the consumption of oxygen and glucose in neurons [2,3], excitotoxicity [4], mitochondrial deficiency, and inflammation [5]. Post-stroke inflammation plays an important role both in the damage development and in the restoration of brain tissue. A special role in this process is played by microglia, which is involved in the immune response immediately after an ischemic stroke and retains its activity up to a month after an ischemic attack [6]. Excessive or inappropriate activation of microglia contributes to chronic inflammation and leads to neuropathological progression [7]. Astrocytes are also involved in the progression and resolution of inflammation after stroke. In a healthy brain, astrocytes maintain homeostasis and help to maintain neuroplasticity [8,9]. After a stroke, these cells can initiate a glial scar formation, and thus inhibit axon regeneration and initiate neurotoxicity. During glial scar formation astrocytes demonstrate hypertrophied interdigitated processes that form a physical barrier, which catastrophically affects the trophism of the nervous tissue. Preservation of the functionality of astrocytes and a reduction in their involvement in the structure of the glial scar can positively affect the safety of neurons and recovery after ischemic stroke [10,11]. 

Thus, increasing the efficacy of rehabilitation measures through adjuvant pharmacotherapy in patients with cerebral disorders is an important challenge [12,13]. Racetam family members, in particular piracetam (Figure 1, Compound **1**), improves the patients rehabilitation time and efficacy. 

The previously synthesized compound (*R,S*)-2-(2-oxo-4-phenylpyrrolidin-1-yl)-acetamide (Figure 1, Compound **2**) turned out to be an effective drug with a wide spectrum of neuroprotective actions (Phenotropil) [14]. Phenotropil shows pronounced antiamnesic activity at doses of 50–400 mg/kg, and a direct effect on the integrative functions of the brain, memory consolidation, attention improvement, and easier learning provision [15]. Moreover, those positive changes in the intellectual–mnestic sphere of patients are accompanied by an improvement in associative–intellectual processes, decreased astheno-depressive syndrome, and an improved mood without an increase in mental stress and euphoria in patients [15,16]. Experiments on laboratory animals show that racetams affect the main neurotransmitter systems of the brain, including cholinergic, dopaminergic, GABAergic, and glutamatergic. Additionally these drugs affect memory processes. Previously, we carried out a chemical modification of Compound **2** by introducing a fragment of the potassium salt of taurine into its composition (Figure 1, Compound **3**) in order to increase the pharmacological efficacy in the treatment of post-stroke state in rats, as well as to obtain its water-soluble form suitable for the injection route of administration [17]. We have shown that Compound **3** better restores motor function and improves the cognitive abilities of animals in the post-stroke period, compared with the well-known water-soluble nootropic of the same class—piracetam (Compound **1**). In order to study the effect of changing the length of the alkyl tail, as well as varying the chemical environment of the lactam fragment on biological activity, in this work we synthesized the compound 3-(2-oxo-4-phenylpyrrolidin-1-yl)propane-1-sulfonate calcium(II) (Figure 1, Compound **4**). It was shown both in vitro and in vivo that homotaurine can exhibit neuroprotective and neurotropic activity through various mechanisms, including effects against oxidative DNA damage, antifibrillogenic activity, as well as antinociceptive and analgesic activities [18,19,20]. Therefore, we hypothesized that the proposed chemical modification would further enhance the nootropic activity. We investigated Compound **4** in in vitro models of inflammation and in in vivo rat models of occlusion of the middle cerebral artery (MCAO). ^1^H and ^13^C NMR spectra of Compound **4** are presented in Appendix A (Appendix A, respectively).

## 2. Results

### 2.1. In Vitro Studies

First, we investigated the influence of Compound **4** on a macrophage-derived cell line RAW 264.7. Macrophages play a crucial role in the development, progression, and resolution of ischemic stroke. By secreting different cytokines, these cells can orchestrate the immune response and skew it towards a more or a less proinflammatory profile. We treated macrophage-derived cell line RAW 264.7 with LPS and measured various cytokine levels in the supernatant along with the gene expression levels of different cytokines, involved in the inflammation. 

#### 2.1.1. Inflammatory Gene Expression and Compound 4

We investigated IL-1 β, iNOS, and COX2 expression levels in RAW 264.7 macrophage-derived cell line treated with different concentrations of Compound **4**. COX2 and iNOS were selected, as they are the main mediators of the inflammation. IL-1β is one of the main proinflammatory cytokines. Figure 1 shows the real-time PCR data for gene expression. 

There was a significant, dose-dependent reduction in IL-1β and significant dose-dependent elevation of COX2 in the cells treated with Compound **4**, compared to cells, treated solely with LPS. Compound **4** also significantly inhibited iNOS expression for 60%, however, this effect was not dose dependent. The Kruskal–Wallis test was used to estimate the significance.

#### 2.1.2. Cytometry Bead Array and the Concentration of Cytokines

The gene expression data were further supported by the measurement of cytokine levels in the supernatant from the same cell line, as the secretion of the cytokines does not always repeat the gene expression patterns. Raw 264.7 cells were treated with LPS, and LPS plus different concentrations of Compound **4** (0.5mg/mL and 0.1mg/mL). Then we collected the supernatant and measured the concentration of IFNγ, IL-2, IL-4, IL-6, and IL-10 in treated, untreated, and intact cell supernatants. IL10 and INFγ levels were similar in all groups (Figure 2). 

IL-2 levels were significantly higher in the group treated with 0.5mg/mL Compound **4**, which supports our hypothesis, that Compound **4** is beneficial for stroke management. It was shown that IL-2 administration improves the resolution of the stroke, by controlling the T-reg levels [21]. IL-2 also reduces demyelination after ischemic stroke by suppressing CD8^+^ T cells [22].

IL-4 levels were also elevated in both groups treated with Compound **4**. This elevation may contribute to a better outcome, by driving the anti-inflammatory response in microglia [23]. 

IL-6 levels were significantly different from the intact cells, but, there was no difference between LPS and LPS and Compound **4** treated groups. This can be due to the specific time limitations of our experiment. An additional test of serum concentrations of IL6 can be useful, as macrophages are not the only source of IL-6 during inflammation. 

### 2.2. Animals and Study Design 

In vitro tests of Compound **4** show anti-inflammatory effects by reducing the production in inflammatory cytokines, as well as influencing the expression of genes responsible for the development of inflammation. This data suggested a further rationale for in vivo experiments. Thus, we investigated the effects of Compound **4** on the development and progression of ischemic stroke in the model of MCAO rats. The study design is shown in Figure 2.

#### 2.2.1. Effects of Compound 4 in Rats with MCAO 

After MCAO rats treated with Compound **4** showed a higher rate of weight gain, this tendency was consistent until day 23 (Figure 3).

#### 2.2.2. Neurological Scale

We investigated the effect of the Compound **4** on the improvement of the neurological deficit, using the modified neurological severity score (mNSS) [24]. mNSS includes several motor (muscle condition and abnormal movements), sensory (visual, tactile, and proprioceptive), reflex, and balance tests. 

A maximal level of neurological deficit was observed on day 3 in rats treated with normal saline and Compound **4**. Rats in the Compound **4** group tended to have better neurological scores and improved neurological deficit reduction rate compared to the normal saline group (*p* = 0.55) (Figure 4). 

#### 2.2.3. MRI Stroke Volume 

After the MCAO introduction, we performed the first MRI to confirm the area and the volume of the introduced stroke. After that, we repeated MRI on day 28 to investigate the influence of Compound **4** on cerebral edema. There was no significant difference in MRI stroke volume between Compound **4** and normal saline groups. Thus, on the first day, the volumes of ischemic stroke lesions were 245.7 ± 38.3 and 259.2 ± 28.2, respectively, and on day 28 were 160.3 ± 32.6 and 176.3 ± 28.3. The reduction rate was also not significant. The examples of MRI images are presented in Figure 5.

#### 2.2.4. Behavior Tests 

We further investigate the influence of Compound **4** in different behavioral tests. These tests provide the assessment of locomotor and cognitive behavior of the rats and indirectly illustrate the dynamics of post-stroke brain injury progression. 

#### Hole-Board Test 

Rats treated with Compound **4** were significantly more active, according to the hole-board test. Thus, the number of sectors crossed was higher in the compound group, and the locomotor activity remained better from day 2 until day 22 (*p* < 0.01). Grooming time was the same for both groups, and the immobilization time had the same pattern as the number of sectors crossed. Animals treated with normal saline had a significantly longer immobility time than animals treated with Compound **4** (Figure 6a). 

#### Open Field Test

We further assessed the tentative exploratory behavior of the animals in an open field test. Rats treated with Compound **4** showed significantly higher exploratory activity. The number of sectors crossed was also significantly higher in OFT in Compound **4** group (*p* < 0.05). This tendency was observed from day 11 until day 23 (Figure 6b). 

In addition to the locomotor activity tests, we conducted a beam walking to assess the motor deficit of the animals. Before MCAO we taught the animals how to pass the beam-walking test, where the animals were exposed to a resurrecting beam. After MCAO on day 13, we counted the number of steps and slippings for the fore and hind limbs. Rats treated with Compound **4** had significantly lower deficit levels of 59.1 ± 13.4 compared to 78.9 ± 22.1 in the saline group for forelimbs (*p* < 0.05). There was no significant difference between the fore and hind limbs (Figure 6c). 

#### 2.2.5. Immunohistochemistry

As astrocytes play a crucial role in the development and stroke resolution, we investigated the number of astrocytes in two different regions. Primary motor cortex, 0.36 mm to the right from Bregma (M1), and caudate putamen (striatum) (CPu) 0.36 mm to the right from Bregma (Figure 7) [25]. 

For this set of experiments, we included an additional piracetam control group, as piracetam shows cognitive and memory benefits in patients with cerebral ischemia. We hypothesized that the number of astrocytes will be negatively correlated with the resolution of the stroke.

There was a statistically significant reduction in the number of astrocytes in the M1 zone in the animals treated with Compound **4** (mean = 123 astrocytes per image (*p* = 0.0006)) (Figure 8d). There was no significant difference between the piracetam group and normal saline (260 astrocytes vs. 260 per image, respectively). Examples of data are presented in Figure 8 (a—CPu and b—M1). There was a significant difference between piracetam and Compound **4** observed also in the CPu area, (*p* = 0.0120) (Figure 8c).

These findings highlight the anti-inflammatory effect of Compound **4** and could explain the cognitive behavior benefits of Compound **4**. 

## 3. Discussion

In cerebral infarction, when blood flow is 10–25% below normal, nerve cells undergo irreversible damage or even death, while inflammatory cells release inflammatory factors [26].

In this study, we investigated both the in vivo and in vitro influence of Compound **4** on inflammation and stroke. As macrophages are one of the main drivers of inflammation during ischemia and stroke, we looked at the polarization of the macrophage-derived cell line under different conditions. We also focused on astrocytes as these cells drive the glial scar formation and can influence the outcomes of the stroke. Additionally, we used an in vivo model of MCAO in rats to investigate the effects of the compound on cognitive and locomotor activities, as the impairment of these functions usually leads to disability in humans. 

According to our results from the model of LPS-induced RAW 264.7 cells, Compound **4** has a pronounced anti-inflammatory effect and influence on gene expression and production in cytokines. Thus, the addition of Compound **4** led to a reduced expression of genes responsible for the production of such pro-inflammatory factors as IL-1b, iNOS, and COX2. Inhibition of IL-1b production prevents the activation of phospholipase A2 by this cytokine. Phospholipase A2 participates in the degradation of arachidonic acid and the destruction of the phospholipid bilayer. This can contribute to an increased microvascular permeability, impaired blood–brain barrier (BBB) function, formation of vasogenic cerebral edema, and reduction in such metabolites as prostaglandin and leukotriene [27]. Furthermore, reduced levels of IL-1b weaken the activation of microglia. As microglia are the main effector cells of the neuroinflammatory response, they can exacerbate the inflammation, which will further lead to secondary brain damage via secretion of several potentially neurotoxic substances, such as TNF-a and iNOS. 

Inflammatory cytokines are increased in ischemic stroke and correlate with worse clinical outcomes in patients and larger infarct size in animal models [28,29]. It is known that the racetam family members have an anti-inflammatory activity. For instance, piracetam protected against inflammatory response caused by an intracerebroventricular injection of LPS into the rats’ brains, reduced inflammation and cell death [30]. According to the results of another research study, piracetam reduced the pain from the inflammatory response due to the inhibition of cytokines production and oxidative stress [31]. Oxiracetam suppressed the activation of BV2 microglial cells, decreased the production of AB-induced inflammatory molecules (expression of cytokines IL-1β, IL-6, and TNF-α) and NO in BV2 cells, and protected hippocampal HT22 cells against indirect BV2 cell-conditioned medium [32]. R-phenylpiraceta suppresses the overexpression of inflammatory genes, such as tumor necrosis factor-α (TNF-α), interleukin 1 beta (IL-1β) and inducible nitric oxide synthase (iNOS). R-phenylpiraceta is also potent against inflammatory gene overexpression, such as tumor necrosis factor-α (TNF-α), interleukin 1 beta (IL-1β) and inducible nitric oxide synthase (iNOS) [33]. In our experiments, treatment with Compound **4** was associated with higher levels of IL-2 and IL-4. Both these cytokines regulate the CD8+ T cell responses and regulatory T cells, leading to anti-inflammatory effects. It was shown that the administration of IL-2 monoclonal antibodies reduced demyelination [22]. Additionally, IL-4 deficiency leads to neurological disfunction and skews the microglia more towards the M1 inflammatory response [22,23,34]. There were no significant differences observed in the IL-10, IL-6, and IFNγ levels, this can be explained by the fact that we investigated the effect of Compound **4** only on the cell line, and further investigation is required. Additionally, there could be some secretion time peculiarities in ischemic stroke that were not visible in our model. 

Thus, according to the in vitro tests, Compound **4** has anti-inflammatory activity and, possibly, can influence the change in the macrophage phenotype from M1 to M2. In turn, it is known that recruited macrophages in the area of ischemic stroke can affect microglia, and microglia can affect macrophages by influencing each other’s phenotype (switching from M2 to M1 phenotype) [35]. The modulation of mutual alternating activation of phenotypes at different stages of ischemic stroke may allow the control of inflammation and neurological recovery after ischemic injury [35]. In addition, by directing the polarization of microglia towards the M2 phenotype, it inhibits the activation of astrocytes and the development of the glial scar [36]. 

Our study showed the presence of a neurological deficit in animals after MCAO.The recovery of the neurological function was faster in rats treated with Compound **4**. Previously, we have demonstrated that the modified Phenotropil (Compound **3**) has neurotropic activity and improves neurological function after stroke. However, the introduction of homotraurine, a nootron substance, which has neuroprojective activity, into the scaffold can improve post-stroke rehabilitation. Such modification was able not only to increase the solubility of the Compound **4**, but also reduce the possible risk of hypertension, compared to its analogue—piracetam. Thus, rats treated with Compound **4** showed higher locomotor and exploratory activity in all behavioral tests we conducted. However, we did not see any differences in the volume of ischemic tissue on MRI, which can be explained by the time of formation of both the focus of necrosis and vasogenic edema. The most commonly used drug in clinical practice in the post-stroke period, Oxiracetam, shows similar results in animal models. Rats treated with Oxiracetam in the post-ischemic period moved more and memorized faster [37]. 

Stroke therapy involves the administration of drugs during the “golden hours” after a vascular accident, while in our study the first administration of the substance occurred 24 h after ischemia was simulated. We started the treatment 24 h after the MCAO induction, as we assumed that most of the patients receive the therapy with around 24 h delay after the stroke. The use of nootropics in the post-stroke period aims to restore the functions of damaged neurons and form new neuronal connections. Nootropics are not supposed to reduce the volume of the stroke lesion. Leviterocetam improves the cognitive functions of rats in the MCAO model by enhancing angiogenesis and reducing the inflammatory process [38]. Microglial activation can persist for a long time in the post-stroke state and impair the recovery of mnestic functions [39]. Moreover, the synthesis of inflammatory cytokines by microglia, such as IL-1β, may promote apoptosis of neurons in the motor zone and the hippocampus. In our opinion, it is this mechanism that explains the restoration of the orienting-exploratory activity of animals in behavioral tests. This is confirmed by a post-mortem reduced number of astrocytes in brain areas in animals treated with Compound **4** (Figure 8).

The effects of Compound **4** in both in vivo and in vitro models illustrate the potential benefits of new *N*-hetero cycle derivative in the management of postischemic patients, as there are very limited treatment options after stroke, especially due to very limited treatment options 24 h after a stroke. The results of our studies indicate the prospects for the synthesis of a library of compounds containing, along with a homotaurine residue, fragments of various lactams in order to enhance the biological activity of this class of compounds.

## 4. Materials and Methods

The purities of Compound **4** for biological testing were assessed by NMR to be 95%. NMR spectra were detected on a Bruker Avance II 300 spectrometer, Germany (300 (^1^H), 75 MHz (^13^C)) in D_2_O in the pulse mode followed by Fourier transform, where Me4Si was used as an internal standard. Spin multiplicities were described as s (singlet), d (doublet), t (triplet), or q (quartet). IR spectra in the solid phase were recorded on a Bruker Tensor-27 instrument with the attenuated total internal reflectance (ATR) module. Elemental analyses were carried out at the Laboratory of Organic Microanalysis of INEOS RAS. 

^1^H NMR (D_2_O, ppm, *J*/Hz): 2.38–2.69 (m, 4H, H-3), 2.59(t, J = 7.1, 4H, (N**CH**_2_CH_2_CH_2_SO_3_)_2_), 3.31–3.40 (m, 2H, NCH_2_CH_2_CH_2_SO_3_), 3.65 (t, J = 7.1, 4H, (NCH_2_CH_2_**CH**_2_SO_3_)_2_), 3.19–3.60 (m, 4H, H-5), 3.66 (m, 2H, H-4), 3.45–3.60 (m, 4H, H-5), 7.09–7.26 (m, 10H, Ph), ^13^C NMR (D_2_O, ppm): 21.99 ((N**CH**_2_CH_2_CH_2_SO_3_)_2_), 41.22 ((NCH_2_**CH**_2_CH_2_SO_3_)_2_), 54.45 ((NCH_2_CH_2_**CH**_2_SO_3_)_2_), 36.45 (C-4), 38.48 (C-3), 48.32 (C-5), 126.74 (2C, C_orto_), 127.11 (C_para_), 128.93 (2C, C_meta_), 142.28 (C_ipso_), 176.78 (C-2). IR (ν/cm^1^): 742, 1044, 1183, 1668 (C=O). Found (%): C, 51.64; H, 5.33; N, 4.63; S 10.60. C_alcd_. (%): C, 51.33; H, 5.47; N, 4.73; S 10.33. 

### 4.1. Cell Culture

A murine macrophage cell line RAW 264.7 were grown in high-glucose DMEM supplemented with 10% FBS, and 1% antibiotics (penicillin and streptomycin) in a humidified incubator flushed continuously at 37 °C with 5% CO_2_. The culture medium was refreshed every 3 days.

Cells (250,000/well) were plated in 6-well tissue culture plates and incubated for 24 h in a complete medium to reattach to the surface. After 24 h, LPS (*Escherichia coli*, Sigma-Aldrich, St. Louis, MI, USA, L4130 0111:B4) at a concentration of 10 μg/mL was added to the cells. After 18 h of incubation, Compound **4** (0.1 and 0.5 mg/mL) was added and incubated for another 6 h. The anti-inflammatory effect of Compound **4** was compared with the effect caused by LPS within 24 h. After 24 h of total incubation of cells with LPS and 6 h of incubation with Compound **4**, the medium was taken for analysis of cytokine production, and the cells were scraped to analyze the expression of target genes.

### 4.2. Cytokine Expression 

The level of cytokines in cell culture supernatants was determined by flow cytometry using a commercial cytometry bid array Mouse Th1/Th2/Th17 CBA Kit (BD Biosciences, San Diego, CA, USA, cat. no. 560485). The sample preparation was carried out following the manufacturer’s protocol. Data collection was performed on a Beckman MoFlo XDP flow cytometer (Beckman Coulter, Miami, FL, USA) with Summit V5.2.0.7477 software installed. The following device configuration was used in the experiment: blue, green, and red lasers with wavelengths of 488 nm, 561 nm, and 628 nm, respectively. The Fl-8 (580/23 nm)/Fl-10 (670/30 nm) channels were used to detect the signal from particles carrying PE-conjugated antibodies. Processing and analysis of the results were carried out using the FCAP Array Software v3.0—BD Biosciences. Mean and standard deviation from different replicas were used to determine the concentration of the cytokines.

### 4.3. Quantitative Real-Time PCR Analysis

Analysis of the level of gene expression was carried out using real-time reverse transcription-polymerase chain reaction (PCR).

Total RNA was isolated from cells using the RNA-Extran kit (Synthol, Moscow, Russia) following the manufacturer’s protocol. The corresponding cDNA (cDNA) was obtained in the course of a reverse transcription reaction using the OT-1 reagent kit (Synthol, Moscow, Russia) and a PTC (100) programmable device (MJ Research Inc., USA). The reaction program included incubation of the mixture for 10 min at 25 °C, 30 min at 39 °C, and 5 min at 92 °C, followed by cooling to 4 °C.

Real-time PCR was performed on a CFX96 Touch™ Real-Time PCR Detection System (Bio-Rad Laboratories, Hercules, CA, USA) in the presence of Eva Green fluorescent dye (Synthol, Moscow, Russia). The amplification program included 3 min at 95 °C, and 40 cycles (10 s 95 °C, 10 s 57 °C, 30 s 72 °C). The composition of the reaction mixture (25 μL): 1x PCR buffer; 8 mM MgCl2; 0.25 mM mixture of deoxynucleotide triphosphates (dNTPs); 0.2 units Taq polymerases; 200 nM forward and reverse primers (Synthol, Moscow, Russia). The primers were selected using the Primer-BLAST (NCBI) program and are presented in Table 1. To determine the change in the expression level, the threshold cycle calculation method was used. Samples were normalized to the amount of mRNA of the Rpl27 gene.

### 4.4. In Vivo Experiments

The animal experiments were carried out in 2 stages: first stage—studies of the behavior with the introduction of Compound **4** on intact animals, to determine the most effective dose. Secondly, we modeled ischemic stroke by the method of occlusion of the middle cerebral artery followed by the intraperitoneal injections of Compound **4**. We investigated the neurological status and locomotor activity of the animals along with monitoring the ischemic stroke lesion size and weight gain. 

Animals used in the experiments were SHR rats of the SPF category. Gender—males; Number of animals—23 animals were obtained from Charles River Laboratories (Germany). All animals were maintained according to the standards of the 2010/63/EU Guidelines on the treatment of animals used for research purposes. The animals were kept in a conventional animal facility of the Pirogov Russian National Research Medical University under an automated day (08:00 to 20:00) and night (20:00 to 08:00) cycle with at least 12-fold exchange of air per hour and optimal temperature and humidity of 20–24 °C and 45–65%, respectively. The experimental protocols were approved by the Pirogov Russian National Research Medical University Animal Care and Use Commission. The animal procedures were of medium severity and caused short-term medium-level pain or stress. 

Before conducting the experiments, the animals underwent the HBT and were allocated to groups based on their behavior score and weight. Kullback–Leibler divergence method for allocation based on two numerical parameters was used for the allocation. Before the experiment, we trained animals to pass the beam-walking test for 3 days. Then we initiated MCAO stroke. Compound **4** (*n* = 11), and saline (*n* = 9) were intravenously injected at 24 h after establishing the stroke model and subsequently daily for the next 5 days. Compound **4** was injected at a dose of 100 mg/kg and an injection volume of 0.3 mL. For IHC, 10 rats were included. Piracetam was also administered intravenously for 5 days after the stroke at a dose of 300 mg/kg. Since piracetam is not a reference drug for the treatment of ischemic stroke in the acute period, rats, under the influence of piracetam, were not tested in behavioral tests but were used for post-mortem evaluation of cellular composition in the brain.

The following predetermined exclusion criteria were used: incomplete MCAO indicated by an incomplete lesion onMRI after 24 h [*n* = 3] and post-operative weight loss of > 20% [*n* = 2], and severe animal condition within the first post-stroke 48 h [*n* = 3]. Animals reached humane endpoints based on the absence of nociceptive reflex withdrawal responses, hypothermia, and heart rate reduction to 100–150 beats min. As steroid treatment could have influenced the experiment, rats were humanely euthanized according to the ethical protocol.

### 4.5. Transient Middle Cerebral Artery Occlusion

Transient Middle Cerebral Artery Occlusion was conducted as previously described [17]. The rats were anesthetized initially using 3% isoflurane and maintained with a mixture of 2–2.5% isoflurane and 97.5–98% atmospheric air (EZ-7000 Classic System, E-Z Anesthesia^®^ Systems). Next, 0.05 mg kg^−1^ atropine sulfate dissolved in 1 mL 0.9% NaCl was injected intraperitoneally to reduce respiratory tract secretion. In addition, 0.1 mL of 0.5% we injected ropivacaine subcutaneously along the prospective incision site (neck midline).

Subsequently, a transient MCAO was conducted for 90 min according to the protocol proposed by Koizumi and modified by Longa. This process was assisted by MRI, so we could visualize the success of the MCAO [40]. During the entire MCAO procedure, the rats were maintained under inhalation anesthesia as mentioned above.

MRI measurements were obtained intraoperatively as previously described and on postoperative days 1, 7, 14, and 28 after operation using a ClinScan (Bruker BioSpin, Billerica, MA, USA) 7T MRI system. We performed T2-weighted imaging to evaluate ischemic lesions (Turbo Spin Echo pulse sequence with restore magnetization pulse and breath synchronization, turbo factor 10, repetition time/echo time 5230/46 ms, averages 2, spectral fat saturation, a field of view 30 21.1 mm, slice thickness 0.7 mm, matrix size 256 162). We analyzed MRI data with ImageJ software [41]. The volume of the hyper intensive ischemic stroke area was calculated by summing the areas of adjacent cross-sections using the following formula: Volume = (S1 + … + Sn) (h + d), where S1, …, Sn are areas measured in n slices, h is the slice thickness, and d is the interval between slices.

### 4.6. Behavior Tests

#### 4.6.1. Hole-Board test

The HBT was performed on postoperative days 7 and 22 after MCAO. Briefly, animals were placed in the middle of the testing arena with regularly arranged holes on the floor. Each animal was observed for 20 min. After every animal, the board was wiped with 70% ethanol and dried. In this test, the following parameters were assessed: orientational and motor activity (horizontal activity, arbitrary units: number of crossings; vertical activity, arbitrary units: number of head dippings); anxiety level (immobility and grooming time and number of grooming and immobility events); and the number of central sector crossings.

#### 4.6.2. Open Field test

The OFT was performed for 3 min on postoperative days 11 and 23. In this test we also measured orientational and motor activity, the number of sectors crossed, and the speed, (horizontal and vertical activities).

Before the study animals were trained to correctly pass the installation. For 3 days, 3 sets daily. Rats were placed in the goal box with nesting materials for 1 min. After 1 min the light and white noise were turned on and the rats were placed on the beam, at the peg hole position closest to the goal box. Once the rat entered the goal box the light and the noise were turned off. Gradually the rats were trained to pass the beam from a further distance from the box. Then the behavior of the animals was assessed using a scoring system, where 1 is a stable balance and 6—animal falls off the beam with no effort to balance. The rest is repeated until the animal learns how to pass the beam and reaches a score of 1 or 2.

Animals that failed to learn the setup were not included in the study (*n* = 1). After modeling ischemia, the test was performed on days 12 and 13. The first day was introductory, the rats remembered the skills of passing the test. On the 13th day the animals were evaluated for the number of steps, slips from the installation, and incorrect limb positions. The performance index for fore and hind limbs was calculated separately using the formula:(1)$$\frac{Error+0.5\times Slip\times 100}{total\;number\;of\;steps}$$

The test was conducted as described in [24]. The test was carried out on the 1st, 3rd, 5th, 7th, and 10th days after the simulation of an ischemic stroke. Shortly several physiologic reflexes are assessed in this test. Each measurement has a score from 0 to 6, with a maximum overall score of 18, corresponding to a severe neurological deficit.

### 4.7. Immunohistochemistry

The brain was removed after perfusion with 4% buffered formalin (Biovitrum, Saint Petersburg, Russia), additionally fixed for 48 h in the same buffered formalin, material: fixative ratio 1:20. After the fixation, a fragment of the ischemic focus was cut out with the capture of intact tissue and placed on a marked histological cassette.

The posting of the obtained material was carried out in an automatic carousel-type processor AGOT-1 (OOO Orion-Medic, Saint Petersburg, Russia) according to the following scheme: 7 changes of isopropyl alcohol (Componentreaktiv, Moscow, Russia) for 45 min. in each, then 3 changes of paraffin (Biovitrum, Saint Petersburg, Russia) for 60 min each, at a temperature of 60 °C. After that, we poured the samples into reusable metal molds with the filling station Leica HistoCore Arcadia (Leica Biosystems, Nussloch, Germany).

The microtomy of the blocks was performed on a manual rotary microtome RM 2125 RTS (Leica Biosystems, Nussloch, Germany), and the sections were transferred to the surface of the water in a 30/60 slide bath for straightening (KB Tekhnom, Ekaterinburg, Russia). The straightened sections were caught on labeled glass with an adhesive coating Snowcoat X-TRA (Leica Biosystems, Nussloch, Germany).

Deparaffinization, unmasking, staining of sections, clarification, and conclusion were performed according to the manufacturer’s method described in the instructions for the Anti-Ig HRP Detection Kits (BD Biosciences, San Diego, CA, USA). Purified Mouse Anti-GFAP cocktail (BD Biosciences, San Diego, CA, USA) was used as primary antibodies. Consul-Mount mounting medium (Thermo Fisher Scientific, Kalamazoo, MI, USA) was used to cap the stained sections. Slides were examined under a bright-field microscope (Carl Zeiss, Jena, Germany, Axioplan 2).

Astrocytes were analyzed with Cell profiler 4.2.1 Broad institute incorporated. The magnification ×20 was used for each image set, and the whole image was analyzed Figure 9A. The first threshold was applied with lower and upper bounds of 0.006 and 0.8 respectively with a method of minimum cross-entropy. With threshold smoothing scale 0 Figure 9B. Then, the threshold image was smoothed with a Gaussian filter Figure 9C. As a third step “Enhanceorsupress” feature was used, and the smoothed image was enhanced with dark holes ranging from 1 to 20 pixels in size Figure 9D. Cells were identified by using the “identify primary object” function with a size range from 8 to 80 pixels. These numbers were manually determined by measuring astrocytes in different areas. New threshold bounds from 0.000001 to 0.004 were used in Figure 9E. The object count was automatically calculated by the software. All objects touching the image border were discarded. The overlay of identified objects with the initial image is presented on Figure 9F.

## Data Availability

The data presented in this study are available in this article.

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
