# Peer review of "The Effect of a New N-hetero Cycle Derivative on Behavior and Inflammation against the Background of Ischemic Stroke"

_molecules, 2022, doi:10.3390/molecules27175488_

Round 1
Reviewer 1 Report
The article by Denis A. Borozdenko et al. entitled “The effect of a new N-heterocycle derivative on behavior and inflammation against the background of ischemic stroke”. The reviewer's enthusiasm remains limited due to the following concerns
1. The author must write the aim of the study in the abstract
2. What is the name of compound 1? Authors must check with assistance from the chemist and authenticate the name of the compound
3. It is not clear what the aim and objectives of the study are in the introduction.
- Figures are not visible and are needed to improve the quality. Authors are required to revise the statistical analysis
5. Significant grammar and typographical errors were found throughout the manuscript. Authors from non-English speaking countries should ensure to have their articles corrected by a native English speaker for grammatical, stylistic, and typographical errors.
6. Discussion has not been written very well.
7. The paper has not been drawn according to the style of MDPI
8. The paper has not been well designed. These findings are very inconclusive and crude in their structure. There is a possibility that most data results are artificial. So, this manuscript is not recommended for publication in this journal.
Author Response
Dear Reviewer! We have read your comments with great pleasure. We thank you for your careful and impartial review of our article.
Below our responses.

Reviewer 2 Report
This study entitled “The effect of a new N-hetero cycle derivative on behavior and inflammation against the background of ischemic stroke” used both RAW 264.7 cell and animal models to discuss the protective effect of compound 1 against ischemic stroke. The results showed that Compound 1 exhibited the excellent therapeutic potentials against ischemic stroke in both RAW 264.7 cells and MCAO-treated rats. However, several major points are raised shown below:
1. All abbreviations used in the abstract should be defined at their first appearance. E.g. lipopolysaccharides (LPS), interleukin-1B (IL-1B)
2. The abstract should be concise. Please delete “The dynamics of the stroke lesion were accessed by MRI”. Only the objectives, the results and conclusions are necessary to be highlighted in the abstract.
3. Although the therapeutic role of taurine has been mentioned in the introduction, what are the reasons that the compound 1 is synthesized in this study ? what is the relationship between taurine and compound 1. The purpose of synthesis of compound 1 should be addressed in the introduction part.
4. In Materials and Methods part, the synthetic methods and identification (e.g. mass spectrometry, NMR….etc) of compound 1 and its purity should be provided.
5. The title “Tests” in line 402 is not clear, please define “Tests” more clearly.
6. Figure 2, “IL-4”, “IL-6” “IL-2” IL-10” “INF-r”. Also, what is “intact”? the cell without treatment with LPS? Usually “control” is used.
7. Figure 3, usually “body weight” is used in the y-axis.
8. In the animal study, why only the saline group and compound 1 group are provided? In this case, how to understand if MCAO successfully induced stroke ? For example, in Figure 4, although the neurological score of compound 1 is lower than the saline group, maybe no significant differences are found among the control and MCAO-treated (saline group, here) rats. Therefore, the results of the control group should be provided in this study.
9. In the discussion part, it is lack of a deep discussion with respect to comparing the therapeutic activity of compound 1 against stroke with other similar compounds or derivatives. Please compare the therapeutic activity of compound 1 with other similar compounds to highlight the importance of this study.
Author Response

(The authors gave the same response as above.)

Round 2
Reviewer 1 Report
The present draft of the paper has been accepted for publication.
Reviewer 2 Report
All questions have been answered appropriately and the manuscript has been revised accordingly. As for the question of point 8, the results of the control group provided in the Supplementary Materials should be fine.